# Research Progress of Quinoa Seeds (*Chenopodium quinoa* Wild.): Nutritional Components, Technological Treatment, and Application

**DOI:** 10.3390/foods12102087

**Published:** 2023-05-22

**Authors:** Hongyan Mu, Sophia Xue, Qingrui Sun, John Shi, Danyang Zhang, Deda Wang, Jianteng Wei

**Affiliations:** 1College of Food Science and Engineering, Qingdao Agricultural University, Qingdao 266109, China; 2Guelph Research and Development Center, Agriculture and Agri-Food Canada, Guelph, ON N1G 5C9, Canada; 3College of Food Science, Heilongjiang Bayi Agricultural University, Daqing 163319, China

**Keywords:** quinoa seeds, nutritional composition, phytochemicals, processing treatment, application

## Abstract

Quinoa (*Chenopodium quinoa* Wild.) is a pseudo-grain that belongs to the amaranth family and has gained attention due to its exceptional nutritional properties. Compared to other grains, quinoa has a higher protein content, a more balanced amino acid profile, unique starch features, higher levels of dietary fiber, and a variety of phytochemicals. In this review, the physicochemical and functional properties of the major nutritional components in quinoa are summarized and compared to those of other grains. Our review also highlights the technological approaches used to improve the quality of quinoa-based products. The challenges of formulating quinoa into food products are addressed, and strategies for overcoming these challenges through technological innovation are discussed. This review also provides examples of common applications of quinoa seeds. Overall, the review underscores the potential benefits of incorporating quinoa into the diet and the importance of developing innovative approaches to enhance the nutritional quality and functionality of quinoa-based products.

## 1. Introduction

Quinoa (*Chenopodium quinoa* Wild.) is a plant species belonging to the Chenopodiaceae family, which originated in the Andes region of South America. Quinoa seeds generally have a small grain size (1.8–2.2 mm) which contains high amounts of protein, lipid, and ash. Their extraordinary adaptability to climate and soil conditions makes it increasingly popular. Presently, quinoa is cultivated widely across the globe, including in Europe, North America, North Africa, and Asia [1,2]. Currently, more than 250 varieties of quinoa seeds are grown, and their chemical and nutritional compositions are greatly impacted by genetic diversity, geographic locations, and cultivating environmental conditions [3]. In the terms of uses, quinoa seeds have been most often used in direct cooking, being incorporated into the food formula as an ingredient in bakery products [4], steamed bread [5], and meat products [6]; in addition, its nutritional components, such as protein, polysaccharides, saponins, etc., are extracted. Quinoa seeds have multiple properties, such as anti-oxidative, anti-inflammatory, immunomodulatory, anticarcinogenic, etc. [7]. The composition of polysaccharides in quinoa seeds is more similar to that of fruits and vegetables; therefore, quinoa polysaccharides are considered prebiotic due to their capability of increasing beneficial bacteria growth [8]. Additionally, quinoa seeds have been proven to be beneficial for reducing obesity [9]. Although quinoa seeds have general characteristics, like cereals do, quinoa is normally considered as a pseudocereal due to its different botanical traits from cereals such as wheat, rye, barley, etc.

This study was undertaken to summarize the up-to-date information about quinoa seeds, including the nutritional composition of quinoa seeds, the innovative processing technologies employed to improve the quality of quinoa-based products, as well as the most current practical applications of quinoa seeds in different sectors. This review’s analysis is expected to provide valuable information and knowledge for further exploring the value and potential of specific characteristics of quinoa seeds. 

## 2. Nutritional Composition of Quinoa

The chemical composition of quinoa seeds has been extensively investigated [10,11,12]. The concentration of nutritional components within quinoa seeds varies depending on the cultivating region (Table 1). Here, the major chemical components, such as protein, starch, dietary fiber, lipids, and phytochemicals, are reviewed by focusing on their physicochemical and functional properties that are different from conventional cereals and the variation with the use of technological treatments. 

### 2.1. Protein

Protein accounts for 11.3–18.9%, with an average contents of 14–15%, in quinoa seeds, which is generally higher than in grains such as oat, barley, rye, and maize, and similar to that in wheat protein (9.85–16.97%) [4,25,26] (Table 2). Depending on the difference in solubility, protein is categorized into albumin, globulin, prolamin, and glutelin. For quinoa seeds, albumin and globulin make up the major proteins, with contents of approximately 35% and 37%, respectively, while the contents of prolamin and glutelin were relatively low, with approximately 9% and 16%, respectively [4]. Prolamin and glutenin are the major proteins found in wheat protein, in the range of 40–50% and 35–40%, respectively. These two types of protein contribute to the formation of gluten networks during dough preparation. Due to the absence of gluten proteins, quinoa seeds have potential for the production of gluten-free food products for patients with celiac disease. Moreover, quinoa seed protein possesses more a balanced amino acid composition, similar to the milk protein of casein [27]. Preliminary research has shown that the content of essential amino acids, especially lysine and sulfur-containing amino acids, are relatively high in quinoa seeds. It has been suggested that quinoa seed protein is highly digestible and biologically available [28,29], and the polypeptides obtained after in vitro simulated gastric digestion of quinoa seed protein exerted excellent antioxidative activity, indicating a positive role against oxidative stress-associated disorders [12].

#### 2.1.1. Functional Properties

The solubility of protein is a very important indicator for evaluating its commercial application value. Proteins with a higher solubility are generally associated with enhanced functional consequence. Cerdán-Leal et al. [30] analyzed the solubility of quinoa seed protein isolate (QPI) at different pH values, and the results showed that improved solubility was achieved when the pH value increased from 4 to 10. The other authors determined that the solubility of QPI ranged between 4.08 and 50.38 g/100 g [31]. The enhanced solubility with the elevating pH value was attributed to the higher pH favoring a variation in protein structure that promotes the exposure of hydrophilic groups. Meanwhile, the ionization of carboxyl and deprotonation of amine groups can provide negatively charged species, resulting in increasing molecular repulsion from agglomeration. These results demonstrated that the native quinoa seed protein is more soluble than cereals [31,32], which might be due to the fact that quinoa protein has a much lower molecular weight. For example, quinoa seed protein contains considerable amounts of fractions with molecular weights less than 50 kDa [4], while wheat proteins are enriched, with molecular weights from 50 to 5000 kDa [33]. 

The solubility of protein is related to various factors, such as the preparation methods, treatment procedures, etc. Spray drying is considered as a satisfying process for achieving protein isolates with better solubility, which is associated with the minimum denaturation of protein because of short processing time [30,34]. Shen et al. [35] compared the effects of drying methods on the solubility of quinoa seed protein, and the results suggested that both of the freeze- and spray-dried quinoa seed protein exhibited higher than 90% solubility, while the vacuum-dried product had the lowest solubility (61%). Although no consistent result has been provided yet, the authors emphasized that the denaturation degree contributes to the final solubility of quinoa seed protein. 

A comparison of the functional properties of protein between three pseudocereals of amaranth, quinoa, and chia was performed by López et al. [36]. The results indicated that QPI exerted a higher water absorption capacity (WAC) than the other two plant proteins, which might be attributed to more hydrophilic amino acids present in the quinoa seed protein. It has been mentioned that the post-treatment procedure after the protein isolation would affect the water-binding capability; for example, different drying methods are related to various structural changes in protein. It suggested that proteins with high WAC may be desirable for food applications, as they restrict water loss and increase the yields of food products.

Limited information is available on the interfacial properties of quinoa seed protein. Preliminary studies showed that a maximum emulsifying capacity was achieved at pH 8 for QPI, no significant difference was observed in the emulsifying activity between black and red colored species, and both fractions showed higher emulsifying activity compared with the other proteins [37]. López-Castejón et al. [38] demonstrated that the quinoa seed protein extracted at pH 9 and 11 showed adequate emulsifying capacity for stabilizing emulsions; furthermore, the protein extracted at pH 9 exhibited greater interfacial activity due to a lower degree of denaturation. In terms of gelling properties, Ruiz et al. [31] found that the heated quinoa seed protein extracted at pH 8 and 9 induced an increasing aggregation and semi-solid gels with dense microstructures, which was explained by the reduced decomposition of protein at these relatively mild alkaline conditions. Contrasting with other proteins such as soy protein isolate, the addition of metal ions, such as calcium ions, did not promote the gel formation of QPI; however, the addition of MgCl_2_ resulted in a decreased gel strength of QPI. The authors ascribed this discrepancy to the different binding properties between proteins and divalent ions. It was demonstrated that quinoa seed protein would more readily form strong gels at acidic pH conditions (pH 3.5), while that extracted at more alkaline conditions (pH 10.5) facilitated a finer, more regular, and more elastic gel structure formation upon heat denaturation, suggesting that the gel formation could be modulated by adjusting different pH values [39].

#### 2.1.2. Extraction Methods

Wet fractionation by alkaline solubilization coupled with isoelectric precipitation is the most important method to obtain higher-purity protein. Föste et al. [40] obtained quinoa seed protein with a yield of 69% by extraction at pH 10 followed by acid precipitation. Guerreo-Ochoa et al. [41] reported that a protein extraction yield of 62% and 76% were obtained at pH 9 and pH 11, respectively. However, a higher degree of quinoa seed protein denaturation occurred at much higher alkaline pH (>10). In view of the increasing interest in intermediate-sized quinoa seed protein for industrial applications, the dry fractionation of quinoa seeds was employed to obtain various fractions with improved nutritional and functional properties. A hybrid dry and aqueous fractionation method coupled with ultrafiltration was used to obtain protein-rich fractions from quinoa seeds, and both the concentration and yield of protein reached more than 60%. Although the purity was lower, the volume of water used was significantly reduced. Additionally, the protein was in its native state, in the form of protein bodies, because of the mild treatment condition. Consequently, the functional property and digestibility were improved [42,43]. 

### 2.2. Carbohydrates

#### 2.2.1. Starch

Starch is the most abundant component in quinoa seeds, with a concentration generally varying in the range of 53.2–73.4% on a dry weight basis. Even though some studies have shown that the concentration of starch in quinoa seeds was less than 32% [44,45], the amylose content in quinoa seeds was reported to range widely in 3.5–22.5% on a dry weight basis [46]. Previous studies have indicated that the amylose of quinoa seeds contains shorter chains than that in barley and adzuki, and the amylopectin is composed of short-chains and very long chains [47]. The starch granules exhibit polygonal or oblong shapes with sizes from 0.4–2.0 µm, much smaller than conventional cereals. Compared to wheat or barley starch, the higher amylopectin content as well as the smaller granular sizes contribute to a much higher paste viscosity, better water-binding capacity, and a higher swelling power. It has been indicated that quinoa seed starch has a lower gelatinization temperature, reduced gel strength, lower retrogradation percentage, and high enzyme accessibility [48,49,50]. A previous study showed that quinoa seed varieties with lower amylose contents tended to have a softer gel texture, which might display good performance in extruded or puffed product preparations [51]. Li et al. [52] observed that the retrogradation proportion (the percentage ratio between the enthalpy change of retrogradation and gelatinization) was comparatively lower than that of wheat, maize, and other grain starches, but higher than that of amaranth starch. The lower retrogradation proportion of quinoa starch might be due to the shorter equivalent chain length of amylopectin. In general, the debranched starches have great potential for food, pharmaceutical, and nutritional applications. The quinoa seed starch might serve as a suitable material for producing debranched starches. However, further research with respect to the debranching characteristics of quinoa starch is needed. 

The intrinsic properties of starch, including the amylose/amylopectin ratio, crystalline structure, size, and shape, play a critical role in the digestibility of starches. Preliminary studies have suggested that higher levels of amylose lowered the starch digestibility. Smaller, polygonal, and rough starch granules exhibited enlarged specific surface areas that facilitated the adsorption of enzymes, and lower crystallinity was conducive to a higher hydrolysis rate [53]. Previous studies have suggested that the amylopectin branch-chain distribution determined the susceptibility of starch granules to enzymatic digestion. Compared to common grain starches such as wheat, corn, sorghum, and millet, quinoa starch is characterized by a higher proportion of short chains with a polymerization degree (DP) of 6–12, which might contribute to the inferior crystallinity and increased digestibility. Additionally, the co-existing components surrounding the starch granules may impact the access of digestive enzymes. Quinoa starch in whole-grain flour was hydrolyzed more rapidly than the isolated starch fraction due to the activity of endogenous digestive enzymes [54]. A similar result was observed by Lu et al. [55], who studied the influence of the interactions of chemical components on the digestibility of quinoa seed starch. The results indicated that the cooked whole-quinoa-seed flour exhibited significantly higher digestibility than the protein and/or lipid-removal fractions. This might explain that the network formed by proteins was weakened because of thermal denaturation, which led to increased interaction between starch molecules to form aggregates. Peng et al. [53] compared the in vitro digestibility of starches from different quinoa varieties. The samples exhibited estimated glycemic index (eGI) values of 86–97, which were in accordance with the aforementioned literature which found that quinoa starch is ready to be digested. However, inclusion of the whole-quinoa flour into wheat bread generated lower digestibility of the product; it was observed that the quinoa starch granules remained relatively intact after baking [4]. 

#### 2.2.2. Dietary Fiber

It has been reported that dietary fibers in quinoa seeds (QDF) are present at varied levels, as well as that dietary fiber makes up 7–16% of quinoa seeds [18]. A study examining of six types of quinoa seed flour showed that the concentration of total dietary fiber ranged from 12.71 to 18.59 g/100 g [14]. The diversified level of dietary fiber in quinoa seeds is determined by multiple factors, including genetics, growth locations, processing treatments, and analytical methods [8]. Based on the solubility, quinoa seed dietary fiber consists of soluble and insoluble fractions. The insoluble dietary fiber accounts for about 80% of the total fiber content, and is mainly constituted of galacturonic acid, arabinose, xylose, glucose, galactose, and rhamnose, while soluble dietary fiber is composed of glucose, galacturonic acid, arabinose, galactose, mannose, and xylose [18]. Another study performed by Chen et al. [56] demonstrated that glucose, arabinose, xylose, and galactose were the primary components of QDF. It was reported that the quinoa seed fiber resembled barley fiber; both had a higher water-holding but lower oil adsorption capacity than the fiber of buckwheat, pea, and mung beans [57]. The physicochemical and functional properties of dietary fibers varied among quinoa seed varieties of different colors. The soluble dietary fibers extracted from the three phenotypes of quinoa (red, black, and white) seeds had higher water-holding and oil-holding capacities; the red quinoa seed variety presented especially stronger gel properties than the white and black phenotypes [58]. The function of dietary fibers has been widely recognized, and increasing research has focused on developing dietary fiber-enriched foods without compromising their sensory or textural characteristics. Quinoa seeds are regarded as a typically healthy food with low GI [59], which might be associated with the hydrolytic enzyme inhibition of dietary fibers. It was reported that quinoa seeds have significantly higher soluble dietary fiber content than conventional cereals, which is conducive for their fermentation as health-promoting components in the colon [58].

### 2.3. Lipids

Determinations and analyses of quinoa seed protein, starch, and phytochemicals have been extensively reported. However, limited information is available on the lipid profile of quinoa seeds. The lipid content in quinoa seeds is 4.0–7.09%, which is comparable with oats, but higher than wheat, barley, and rye (Table 2). It has been reported that the neutral lipids in quinoa seeds account for approximately 40–76.2% of the total lipids, and triacylglycerols represent the major fraction. The polar lipids range from 12.7% to 44.4% [60]. A previous study indicated that quinoa seed lipids varied in their concentrations when using different extraction methods. Przybylski et al. [61] reported that the concentration of total lipids extracted by water saturated with n-butanol was higher than that extracted using diethyl ether, suggesting that more polar lipids are present in quinoa seeds. It has been indicated that quinoa fatty acids are primarily composed of monounsaturated and polyunsaturated fatty acids, with proportions of approximately 27% and 55%, respectively; palmitic acid, oleic, linoleic, and linolenic acids comprise the major fatty acids [62]. A factor analysis on the phytochemical fingerprinting of 28 quinoa seed varieties showed that the fatty acid compositions varied greatly in their contents of palmitic acid and long-chain fatty acids [23]. 

### 2.4. Phytochemicals

#### 2.4.1. Polyphenolic Compounds

Extensive studies have shown that quinoa seed oil is a good source of phytochemicals, in particular of phenolics, phytosterols, and squalene [63,64]. The level of phenolic compounds in quinoa seeds varies greatly corresponding to the cultivation area (Table 1), which bears a similar range to the commonly cultivated grains (Table 2). Phenolic compounds are present in both free and bound forms, and the content of total phenolic compounds varies depending on the colored quinoa seed genotype. Among the phenolic compounds, vanillic acid, ferulic acid, and their derivatives make up the main phenolic acids, while quercetin, kaempferol, and their glycosides constitute the main flavonoids [1]. The contents of phenolic compounds and flavonoids present in quinoa seeds are shown in Table 2. More than 23 phenolic compounds in quinoa seeds were determined in the previous research, of which phenolic acid and flavonoids were the most abundant ones identified. The commonly present phenolic acids and flavonoids are shown in Figure 1 [1]. 

The phenolic compounds are mainly present in the outer layer of the seed coat; therefore, the removal of saponins by pearling treatment generally causes the loss of phenolic compounds. Decreases of 21.5% and 35.2% for free and bound phenolic compounds in quinoa seeds was observed after a 30% pearling degree, which were appreciably lower than those of other cereals. The author explained that phenolic compounds in quinoa seeds might distribute more evenly within the seeds [24]. 

It is noticeable that the bound phenolics remain their original states within the gastrointestinal environment, which indicates that the bound phenolics are covalently conjugated to the moieties of the cell wall structural components. However, the conjugated phenolic compound could be released via acid or alkaline or enzymatic hydrolysis, and the released compounds are bioavailable in the colon and good for colon health [65,66]. An investigation of the extrusion technique was performed by Song et al. [66] by releasing bound polyphenolics within the tested extrusion temperature. The results showed that the level of bound phenolics decreased from 155 mg/kg to 77–84 mg/kg, but the free phenolics doubled compared to the original. The pretreatment processes have shown the impact on the retention of polyphenols in quinoa seeds; different cooking methods cause variation in the composition of polyphenols. Zhang et al. [67] analyzed the composition of polyphenols of black quinoa seeds cooked by different cooking methods. Compared with boiling and roasting, microwave treatment was the most effective method in releasing polyphenols from the grain seeds, which was associated with the highest antioxidant activity and stronger inhibitory effects on α-glucosidase, and finally resulted in delayed starch digestion. Due to the enrichment of phenolic compounds in the seeds, quinoa seeds are increasingly added as ingredients in the production of cereal-based foods, which exert high levels of phenolic compounds and demonstrate improved antioxidative properties [68].

#### 2.4.2. Tocopherols

The tocopherols content in quinoa seeds differed prominently among the included studies. Tang et al. [20] reported that the total tocopherols amounted to approximately 37.49–59.82 µg/g (based on dry weight). Data obtained from the USDA showed that the content of total tocopherols reached over 7 mg/100 g, which was remarkably higher than those of durum wheat, white rice, and yellow corn [68]. Pereira et al. [21] analyzed 39 quinoa samples and showed that the total tocopherol values ranged from 971 µg/100 g to 1764 µg/100 g (Table 1). Although α-tocopherol predominates in most cereals, γ-tocopherol represents the most predominant fraction in quinoa seeds. The levels of tocopherols in some grain and quinoa seeds are shown in Table 2. 

Tocopherols are considered as one of the most important naturally occurring antioxidants, as they possess remarkable protection against undesirable lipid oxidative processes. Supercritical fluid extraction has been proven to be a favorable technique for extracting tocopherols from quinoa seeds; by this means, the tocopherols were more than four times as concentrated in the quinoa seed oil than those obtained by hexane extraction [69]. It has been reported that a saponin-free quinoa ethanol extract displayed preventative properties against fish oil oxidation. Quinoa seed extracts were further employed as coating materials, and it was shown that this could act as a novel glazing agent to enhance the rancidity stability of frozen fatty fish [70,71]. However, the authors did not provide detailed information on the antioxidants. Considering the loss of phenolic antioxidants during the saponin removal process, the oxidative stability of the quinoa seed foods might result from the lipophilic antioxidants.

#### 2.4.3. Phytosterols

The information on phytosterols in quinoa seeds is relatively limited, since a few varieties have been characterized. Ryan et al. [62] analyzed the presence of phytosterols in the quinoa seeds, and the total content of phytosterol was around 80 mg/100 g. Among those phytosterols, β-sitosterol was the most abundant, followed by stigmasterol and campesterol. A similar result was achieved by Chen et al. [23], who analyzed the phytosterol composition from 28 quinoa varieties; their result showed that the contents of phytosterols ranged from lower than 35 mg/g to higher than 45 mg/g. The author suggested that the content of phytosterols was positively associated with the content of linoleic acid, while negatively related to saturated and monounsaturated fatty acids. Recently, Schlag et al. [22] analyzed phytosterols from 34 different quinoa seed accessions using gas chromatography coupled with mass spectrometry in selected ion monitoring mode (GC/MS-SIM), and 20 different sterols were identified with total contents ranging between 120 and 180 mg/100 g. Noticeably, the evaluated samples were all dominated by Δ7-sitosterol, which was quite uncommon in the grain samples. 

#### 2.4.4. Saponins

Saponins of quinoa seeds are a complex mixture of triterpene glycoside derivatives, including oleanolic acid, hederagenin, phytolaccagenic acid, serjanic acid, and 3b,23,30-trihydroxyolean-12-en-28-oic acid. All compounds have hydroxyl and carboxylate groups at C-3 and C-28, respectively. Considerable amounts of saponins have been determined in quinoa seeds, with levels of 2–240 mg/100 g, though little information is available on the content of saponins in wheat, oats, barley, etc. (Table 2). Quinoa seed saponins are considered as antinutritional ingredients, which are mainly present in the pericarps of quinoa seeds. The presence of saponins gives quinoa seeds a bitter taste, and some toxic effects could also occur. However, the influence of saponins should not be neglected when considering the health benefits of quinoa seeds. Saponins have some pharmaceutical potential, such as anti-inflammatory and anti-cancer effects, in addition to reducing cholesterol levels [72,73]. 

To date, about 40 varieties of saponins have been identified in quinoa seeds, which structurally belong to 8 groups (Figure 2) [74]. Saponins are generally removed from the seed coat by using physical dehulling and washing. Additionally, genetic methods may play a role in controlling the production of saponins [24,74]. 

## 3. Technological Approaches in Improving the Potential of Quinoa Seeds

With growing interest in the nutritional and functional benefits of quinoa seeds, the technological methods for producing quinoa seeds, concentrating their functional components, and incorporating them into health-beneficial food has been widely explored. The processing of quinoa seeds plays a critical role in guaranteeing their palatability, digestibility, and nutrient bioavailability. It also helps to improve the physicochemical properties, which greatly influence the texture and sensory quality of the final product. Two categories of treatment methods can be classified based on whether they are accompanied by heat energy input during processing. The characteristics of these methods are summarized in Table 3.

### 3.1. Thermal Treatment

#### 3.1.1. Extrusion

Extrusion is considered as a versatile technique that facilitates food manufacturing with diverse textures and shapes. It is especially used for processing instant foods with crisp structures. The extrusion technique has been widely used as an efficient thermal method to modify the physicochemical properties of cereals. The contents of protein and lipids were reduced after extrusion of quinoa seed flour because of the formation of protein–lipid or starch–lipid complexes; additionally, high temperature induced protein denaturation may lead to decreased solubility. Remarkably higher water absorption and water solubility resulted from the extrusion treatment. The lipid oxidation was promoted due to the degradation of antioxidants at higher extrusion temperatures [75]. A previous study indicated that extruded quinoa seed flour had a relatively low expansion ratio, which suggests that quinoa seeds are suitable for developing food for which no direction expansion is desired [76].

#### 3.1.2. Drying

Drying is a commonly used approach for improving the storage stability of cereals. Various drying methods can lead to a reduction in water activity in the food matrix, thus leading to the inhibition of microbial activity and endogenous enzyme-induced biological reactions. Among these drying methods, microwave treatment is one of the most efficient. The intensity of the microwave was shown to impact the physicochemical properties of the treated quinoa samples. Previous reports have shown that moderate microwave treatment enables quinoa seed starch to obtain a more uniform crystallinity. However, excess treatment caused the opposite effect: increasing the microwave time led to complete gelatinization of the quinoa seed starch. The study found that quinoa seeds treated at 9 W/g power density for 20 s showed moderate stickiness and good quality [78]. Sharma et al. [81] investigated the effects of various types of thermal processing on the bioactive properties of quinoa seed flour. Excluding boiling treatment, the roasting, microwaving, and autoclaving methods positively promoted the release of total phenolic compounds and flavonoids. The microwave-treated flour exhibited maximum antioxidant properties, followed by the flour resulting from roasting and autoclaving. The results suggest that dry heating is more favorable for obtaining quinoa seed products with improved antioxidative functions, whereas moist heating is more effective in inducing net reductions in saponin. Microwaving especially prevailed in terms of the improved functionality of quinoa seeds.

An attempt to prepare powdered quinoa seed flour using spray drying was reported by Romano et al. [78]. The samples exposed to spray drying conditions exhibited higher antioxidant retention and antioxidative activity, and a positive influence on the stability of lipids was observed during the process. Meanwhile, the enhancement of swelling and the moderate agglomeration of quinoa seeds during the drying process endowed the final products with encapsulation properties.

#### 3.1.3. Heating under Pressure

Heating under pressure (HUP) treatment is favorable for maintaining the compositional profile of quinoa seeds in the development of quinoa-based products [79]. Water conditioning followed by heating under 0.36 MPa was used to treat quinoa seed grains by Wu et al. [80], and compared with the boiled, baked, or extruded quinoa seeds, a higher level of retention of active phytochemicals and antioxidation activity was detected. The results also suggest that the treated quinoa seeds achieved a higher retention of antioxidant capacity within the food, since the presence of antioxidant contributes delayed the deterioration of food caused by lipid oxidation.

Chen et al. [86] prepared instant quinoa seeds using different cooking methods, and the results showed that the pressure-cooked seeds possessed the highest porosity and were the most rehydratable. The rehydrated instant quinoa seed products, with softer and thicker textures, were obtained by pressure cooking and microwave cooking. On the other hand, infrared-assisted lyophilization enhanced the products with a stronger retention of flavor, which was explained by the fact that the shorter drying time required by this method led to less of a loss of volatile compounds. Therefore, pressure cooking and infrared-assisted freeze-drying may be applicable in developing instant or dehydrated quinoa-based food products.

### 3.2. Nonthermal Treatment

#### 3.2.1. High Hydrostatic Pressure (HHP)

The HHP is a typical non-thermal processing technique that has been used for food preservation and production, with minimal nutritional loss and chemical composition variation value [82,83]. The high temperature could generate reduced compactness and roughness of the granules. It has been reported that increasing the pressure from 0.1 to 600 MPa causes significant changes in the structural and functional properties of quinoa seed flour. The antioxidative activity was improved due to the promotion of diffusion of extractable phenolic fractions [87,88,89]. Li et al. [89] compared the effects of the HHP on the super-molecular structure and functional properties of quinoa seed starch and maize starch. The results indicated that the degree of crystallinity in the quinoa seed starch dropped from 23.3% to 14.1, as the pressure was up to 500 MPa, while the maize starch remained relatively constant in its crystallinity, suggesting that quinoa seed starch is more susceptible to pressurization. Other authors [90] reported that the water solubility and swelling capacity of the quinoa seed starch treated by the HHP (0–500 MPa) decreased significantly from 44.10% and 22.40 (g/g) to 12.55% and 11.17 (g/g), respectively. It was explained that the HHP treated starch granules could retain their intact structures or partially disintegrate, resulting in limited amylose leaching. The complete gelatinization occurred at 600 MPa, as reflected by the decreased relative crystallinity and the disappearance of the endothermic and lamellar peaks. However, the retrogradation and textural properties of both starch gels were less affected.

#### 3.2.2. Atmospheric Pressure Cold Plasma (ACP)

The ACP is considered as a non-thermal technique with potential antimicrobial efficiency. Extensive studies have found that the rheological, thermal, hydration, and morphological characteristics of quinoa seed flour are greatly impacted upon ACP treatments, and that this effect is significantly dependent on the exposure time and voltage. Water-related properties such as the water-holding capacity, water absorption index, and water solution index decreased with increasing treatment time and strengthened voltage. The treated quinoa seed flour exhibited improved thermal stability and decreased gel strength [84], which was contrary to the result of the aforementioned HHP treated samples [90]. This discrepancy may be related to the different treatment mechanisms, as high energetic ions of cold plasma induce the depolymerization of starches, resulting in smaller fragments [91]. The molecular structure of the starch samples treated by HHP was greatly related to their botanical origin [92].

#### 3.2.3. Sonication

Plant proteins are inferior to animal-originating proteins, since plant proteins tend to have lower solubility, resulting in limited functional properties. Sonication is an outstanding approach in the modification of plant proteins in order to enhance their techno-functionalities. The mechanism of sonication originates from cavitation effects, which generate high shear forces, micro-jetting, and shockwaves, leading to the partial unfolding of the protein structure and the disintegration of large aggregates [93,94]. A previous study indicated that increasing the sonication time could improve the functional properties of QPI [85]. In a very recent study by Luo et al. [95], 1% of quinoa seed protein isolate dispersions at different pH values (5, 7, and 9) were sonicated at 20 kHz, and14.4 W for 5/15 min. The solubility of QPI at neutral and alkaline pHs was significantly increased after sonication treatments. Moreover, compared to the less than 40% solubility of the untreated samples at pH 9, over 80% of the treated QPI was soluble at this pH value. The ultrasonication process was effective in producing quinoa seed protein nanoparticles; the sonicated quinoa seed protein nanoparticles (QPN) had significantly reduced particle sizes and exerted better emulsification efficiency. The high internal phase emulsion stabilized by the QPN had an oil volume ratio up to 0.89 [96,97,98].

## 4. Various Applications

### 4.1. Bakery Food

Quinoa seeds can be used as an alternative bakery ingredient to improve the nutritional value of bakery products. Partial replacement of wheat flour with whole-quinoa-seed flour can produce dough with enhanced nutritional properties, since quinoa seeds have a higher protein content, a more balanced amino acid composition, and a higher content of dietary fibers and phytochemicals. The exceptionally lower gluten protein proportion and reduced starch digestibility make quinoa seeds as a potential material for the development of gluten-free bakery products or foods with low GI that are beneficial for customer health and wellness. The addition of quinoa seed flour into wheat flour produced weaker doughs, and the starch digestibility of bread with quinoa seed flour incorporated was reduced, which was associated with the inhibited digestive enzyme activity in the presence of the polyphenols and dietary fibers in quinoa seed flour [99,100]. Similar results were also noted: that the incorporation of quinoa seed flour contributed to a slower digestion rate of rapidly digested starch and reduced digestion extent of slowly digested fractions [4].

When incorporating quinoa seeds into food processing, it is critical to consider the physical properties and accessibility of the final products. Water-related characteristics, including water absorption, solubility, and swelling ability, are very important indices for evaluating the overall quality of quinoa seed-based products. Moreover, both the smaller granular size and lower content of amylose in quinoa seed starch, as well as the higher level of amphiphilic protein in quinoa seed flour, resulted in higher water absorption capacity and swelling power, and therefore exerted a positive influence on the quality of quinoa–wheat complex products [1,5,24]. The dough formed with the incorporation of quinoa seed flour caused different physical and textural properties compared to wheat flour dough, and resulted in baked products with lower loaf volumes, higher firmness, and denser structures. A previous study found that the addition of 25% quinoa seed flour into wheat flour did not reduce the acceptability of the bread; moreover, a more desirable flavor was denoted [101].

It should be mentioned that quinoa seed flour incorporated into bakery products tends to have decreased specific volume and increased hardness [102,103]. Previous research has indicated that quinoa seed starch has a good air incorporation ability during dough mixing, but the gas retention capability was greatly diminished during baking process [104]. Fermented cereals play a positive role in the organoleptic, nutritional, and shelf-life properties of bakery products [105,106]. Adding fermented quinoa seed flour into bread dough could increase the specific volume and, thus, decrease the crumbliness in comparison with dough made with unfermented flour [107]. A replacement of rice flour and potato starch with quinoa seed flour resulted in significantly reduced hardness and slightly improved springiness of cakes. The study suggested that a 50% quinoa seed flour formulation was favorable for better sensory quality of the final products [108]. The decrease in hardness in quinoa seed-based bakery foods because of the higher levels of fat and dietary fiber and the lack of gluten in quinoa seeds was favorable for the formation of soft and tender dough; however, inconsistent observations have been demonstrated elsewhere [102].

It could be concluded that the addition of quinoa seed flour into dough formulations has different influences on the textural properties and overall accessibility of the cooked products, which may be the result of the different reference samples chosen for the product. In comparison with the wheat flour-based products, increasing the amount of quinoa flour caused an increase in hardness. In cases of corn or rice-based snack foods, the addition of quinoa seed flour enhanced the tenderness of the final products. A previous study indicated that quinoa seed starch, characterized by smaller starch granules and lower amylose content, had higher enzymatic susceptibility, and amylase and proteinase might play a positive role to obtain increased volume in gluten-free products [109,110]. A gluten-free bread formulated with 50% quinoa seed flour, 38% corn/potato starch, and 15% whey protein isolate/sodium caseinate was produced with significantly increased specific volume. Similar results demonstrated that the tomographic attributes, such as the pore-sized distribution and wall thickness surrounding the pores, were associated with the hardness of extruded cereal-based snacks [103].

### 4.2. Meat Analogues

The exploration of plant-based meat products to replace meat sources is currently gaining remarkably increasing interest because of the appeal of human health maintenance, environmental protection, and sustainability. In case of their complete nutrient profile and bioactive properties, quinoa seeds have been incorporated into meat products either as ingredients or as fat alternatives [111]. Quinoa seed flour and quinoa seed starch have been added into the meat formula to improve the quality of the final meat product based on a health point view. Applications of quinoa seed flour commonly use at concentrations of less than 15%. One study found that quinoa seed flour has no negative effects on the physicochemical, textural, or sensory properties of the meat products; additionally, a specific amount of quinoa seed flour incorporated into the products induced improved oxidative stability [112,113,114]. Felix et al. [115] studied the effects of concentration and temperature on the rheological properties of quinoa seed flour-based gels, and it was indicated that quinoa seed flour in excess of 200 g/kg would facilitate the formation of gels with sausage-like textures, which might have potential in the design of meat-free products catering to vegan diets [115]. Quinoa seeds and starch have been formulated as ingredients to prepare chicken meatballs. Quinoa seed starch was more effective in hindering water loss during cooking and repeated freeze–thaw processes, and the best acceptability was observed with the quinoa seed-starch co-incorporated products [116]. In addition, compared with quinoa seed flour, the patties with the addition of quinoa seeds had lower cooking yields, attributed to the difficulty which quinoa seed components have in interacting with the meat matrix [6].

### 4.3. Plant Milk

Plant-based food development has received rising interest within recent decades. The plant-based food market has become one of the fastest-growing sectors of the modern food industry. The revenue from plant-based protein is predicted to reach over USD 35 by 2024. Plant milk is regarded as a prevalent cow milk alternative. Among plant milk, soy milk is the most popular due to its higher protein content. However, the presence of undesirable components may limit its wide consumption [117]. Despite the lower allergenic capacities of rice milk, the lower protein content and unbalanced amino acid constitution do not make it fully nutritional options [118]. Quinoa seeds have a relatively high protein content, and quinoa protein contains higher ratios of lysine (2.4–7.8 g/100 g protein), methionine (0.3–9.1 g/100 g protein), and threonine (2.1–8.9 g/100 g protein), which are generally the limitations of amino acids in cereals [12]. A previous study indicated that the GI of quinoa seed milk was at a significantly lower level than that of rice milk (52 vs. 79), suggesting that quinoa seeds might be a raw material able to satisfy sensory quality requirements [119].

### 4.4. Fermented Beverages

As documented in the Food and Agriculture Organization of the UN, quinoa seeds have been listed as one of the most popular pseudocereals for brewing. Un-malted quinoa seed and quinoa flakes are employed for brewing beer. Research results have shown that, despite the reduced soluble nitrogen content, the foam stability was significantly higher with 30% quinoa seed flakes, which can be explained by that fact that the higher levels of soluble proteins in quinoa seeds generate high molecular nitrogenous components. When quinoa seed flakes are incorporated in beer, they contribute to a higher sensory desirability than all-malt beer [120].

The potential of quinoa seed protein for wine phenolic fining was evaluated by Pino-Ramos et al. [121]. Compared to commercial fining agents, quinoa seed protein at the concentration of 30 g/hL or 50 g/hL had a similar efficacy in reducing the turbidity of red wines. The study suggested that quinoa seed protein may act as an alternative to animal protein in the wine fining process.

### 4.5. Delivering Hydrophobic Bioactive Agents

The construction of biopolymer-based systems for delivering functional ingredients is intended to improve their solubility, stability, bioaccessibility, and bioavailability, and has become one of the most recent research hotspots. A wide range of proteins and polysaccharides have been reported to play critical roles in this issue because of their biodegradability and biocompatibility. Quinoa seed proteins could form nano-micelles as delivery carriers for hydrophobic compounds by forming amorphous complexes. The encapsulation efficiency and loading capacity of quercetin amounted to 81.3% and 33.9%, respectively. Moreover, the system provided the hydrophobic agents with high retention after 3 months of storage [122]. The distinct characteristics of quinoa seed starch, with their smaller granular size and comparatively higher amount of amylopectin unit chains, allowed it to form a physical barrier for stabilizing emulsion droplets [123]. Quinoa seed starch nanoparticles exerted higher efficiency than maize starch nanoparticles in terms of loading quercetin, which was attributed to the lower crystallinity induced by the high amylopectin content in quinoa seeds, resulting in a higher quercetin adsorption. The stability of quercetin was significantly enhanced, and the incorporation of quercetin significantly delayed the enzymatic hydrolysis of starch nanoparticles [124]. Similar results were observed in rutin-loaded starch nanoparticles with enhanced encapsulation efficiency and loading capacity compared to maize starch nanoparticles [125]. Moreover, the octenyl succinic anhydride-modified quinoa seed starch showed an even better entrapping ability. The encapsulation efficiency of rutin-loaded Pickering emulsions stabilized by amphiphilic quinoa starch reached as high as 99.3%, and sustained release was achieved in the in vitro studies [126]. Previous studies have indicated that microspheres with high adsorption capacity could be fabricated from quinoa seed starch, and could act as a promising material for drug delivery [127].

### 4.6. Edible Films

Quinoa seeds are potential alternative film materials contributing to the dedication to environmental friendliness and development of new food products. Mixed films incorporating quinoa seed protein isolate combined together with chitosan showed enhanced mechanical properties compared with chitosan films, indicating a synergistic effect of the two biopolymers [128]. Previous studies have reported that native quinoa seed starch could be used to prepare transparent biodegradable films [129], whereas many kinds of botanical starches generally require modification for their applications.

## 5. Conclusions and Prospects

This article reviewed the physicochemical and functional properties of the major nutritional components of quinoa seeds, including protein, polysaccharides, lipids, and phytochemicals, based on emphasizing their unique features that differ from those of conventional cereals. The recent treatment techniques, both thermal and non-thermal, employed in processing of quinoa seeds play crucial roles in the acceptability and accessibility of quinoa seed-based products. In addition, the nutritional and functional properties of the final products are affected by the processing methods and conditions. The most current applications of quinoa seeds in the food industry are bakery products, meat analogs, fermented beverages, plant milk, and edible films. Quinoa seed proteins and polysaccharides are regarded as having potential for designing delivery systems.

So far, the development of quinoa seeds for commercial applications is still limited due to the high cost of the seeds. However, quinoa seeds can still be considered as a promising material contributing to environmental protection and sustainability. It can be postulated that with the advancement of technology in cultivation and processing, this novel grain source will be more available and more extensively investigated. The consumption of quinoa seeds as a staple food or a major food ingredient with improved sensory characteristics and universal acceptability remains a focus in food development, and could benefit from the modification of the physicochemical properties of nutritional components via innovative technological approaches. The development of gluten-free and low-GI products may be highly valuable for the maintenance of human health. Additionally, the unique features of quinoa seed proteins and starch, including a higher protein value, a complete amino acid profile, more short chains of amylose, and, especially, longer chains of amylopectin configuration, could make quinoa seeds a potential delivery matrix for functional components.

## Figures and Tables

**Figure 1 foods-12-02087-f001:**
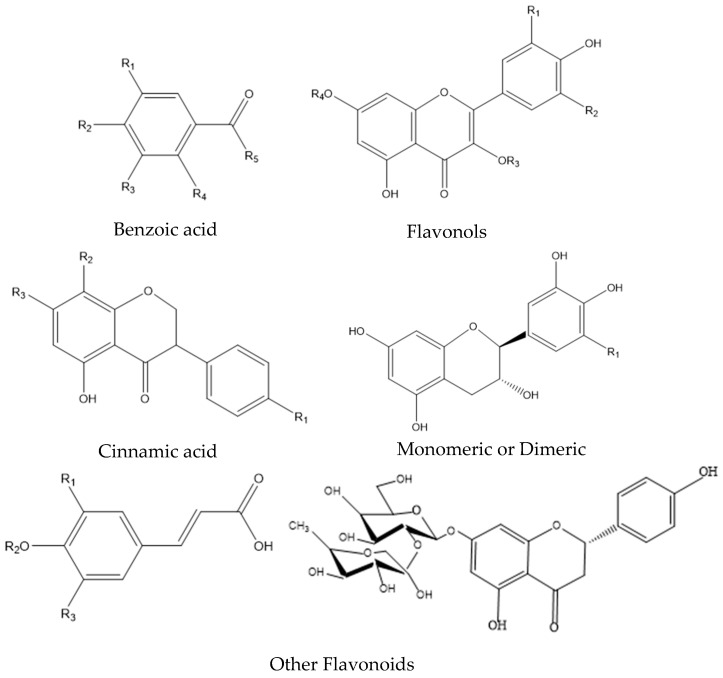
Groups of phenolic compounds found in quinoa seeds (Modified from Tang et al. [1]).

**Figure 2 foods-12-02087-f002:**
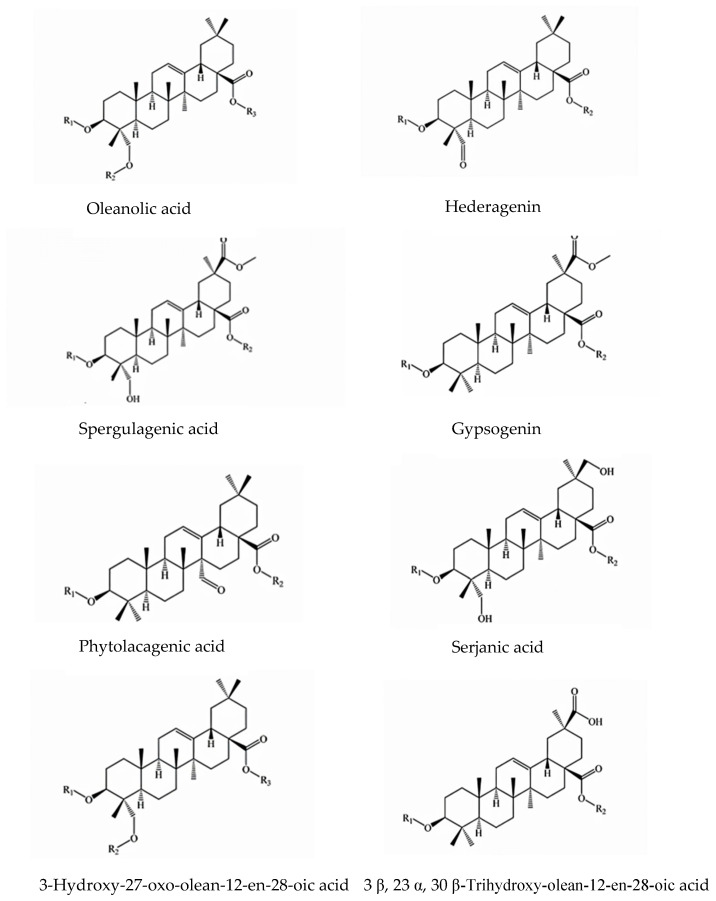
Classification of triterpene saponins isolated from *C. quinoa* Wild. [66].

**Table 1 foods-12-02087-t001:** The major nutritional and functional components found in quinoa seeds.

Nutritional Components	Content	Unit	Origin	Reference
Protein	11.3–14.7	g/100 g	Peru	[13]
	11.6–13.7	g/100 g	Europe	[14]
	14.7–18.9	g/100 g	China	
Starch	55.6–63.0	g/100 g	China	
	53.2–61.3	g/100 g	Peru	[15]
	53.2–73.4	g/100 g	U.S.A	[16]
Dietary fiber	13.66–16.0	g/100 g	Peru	[17]
	7.7–15.9	g/100 g	Spain and the Andean	[3,18]
	12.71–18.59	g/100 g	Europe	[14]
Lipids	4.0–6.9	g/100 g	Peru	[13]
	4.9–6.5	g/100 g	Europe	[14]
	4.11–7.09	g/100 g	China	
Phenolic compounds	30.3–59.7	mg/100 g	Peru	[13]
	46.7–68.2	mg/100 g	Canada	[1]
	75.3–87.58	mg/100 g	Europe	[14]
	51.5–141.95	mg/100 g	China	
	66–202	mg/100 g	Africa, Egypt	[19]
Flavonoids	36.2–72.6	mg/100 g	Peru	[13]
	127–288.8	mg/100 g	Africa, Egypt	[19]
	175–400	mg/100 g	China	[19]
	89.7–93.45	mg/100 g	Europe, Serbia	[19]
Tocopherols	37.5–59.8	µg/g	South America	[20]
	971–1764	µg/100 g	Peru	[21]
Phytosterols	120–180	mg/100 g	N.A.	[22]
	28.7–67.7	mg/g	U.S.A, South America	[23]
Saponins	244.3	mg/100 g	Europe	[24]
	2.76–4.12	mg/100 g	Africa, Egypt	[19]
	15.50	mg/100 g	China	[19]

**Table 2 foods-12-02087-t002:** Comparison of the chemical composition of grains and quinoa seeds.

Composition	Quinoa	Wheat	Maize	Oats	Rye	Barley	Buckwheat	Rice
Protein (g/100 g)	11.3–18.9	10–18	7.8–11.2	12.1–14.1	10.8–12.7	10.8–13.6	11–17	7–8
Starch (g/100 g)	53.2–73.4	60–75	66–78	41–53	63–66	56.7–64.3	60–70	70–80
Dietary fiber (g/100 g)	7.7–18.59	1.14	1.4–2.2	8.8–13.4	1.5–2.0	3.5–5.4	3.4–6.5	0.2–0.9
Lipids (g/100 g)	4.0–7.09	2–2.5	4.1–12.3	4.4–7.2	1.7–2.1	2.4–3.4	2–3	1.3–1.8
Phenolics (mg/100 g)	30.3–202	46–134	60–460	320	136	45–135	70.4–124	48–467
Flavonoids (mg/100 g)	36.2–288	102	260	380	116.7	62–300.8	387	13.49–169.22
Tocopherol (µg/100 g)	0.4–1700	0–340	669–1300	721	68.4–290	747	0.1–8.51	70–190
Phytosterols (mg/100 g)	0.3–180	70–92	43.6	35–49	95.5	50.4	60	13.62–52.71
Saponins (mg/100 g)	2–244	-	-	0.02–0.05	-	-	-	-

**Table 3 foods-12-02087-t003:** Technological approaches for improving the potential of quinoa seeds.

Potential Treatment Techniques for Quinoa Seeds and Their Products		Mechanism Involved	Characteristics of the Treated Quinoa	Reference
Thermal treatment	Extrusion	Heat, mechanical energy, and pressure provided by screw extrusion; the starch in quinoa seeds gelatinizes and the protein denatures, thus changing the structure and nutritional characteristics of quinoa seeds.	(1) Moderate expansion; (2) The original structures of the saponins are destroyed into smaller fragments; (3) Formation of protein/starch–lipid complexes is induced; (4) Degradation of phenolic compounds; (5) Improved protein digestibility due to denaturation.	[75,76]
	Drying	Removes most of the free water in quinoa seeds by the action of high thermal energy.	(1) Improved stability of quinoa seeds during storage; (2) Enhanced antioxidant compounds retention and improved antioxidative properties; (3) Increased swelling.	[77,78]
	Heating under pressure	High temperatures and high pressure provided by high-pressure-resistant equipment changes the structure and nutritional characteristics of quinoa seeds.	(1) Higher levels of active phytochemicals and antioxidants can be retained; (2) Quinoa seed products have better rehydration capacity.	[79,80,81]
Non-thermal treatment	High hydrostatic pressure (HHP)	Water or other fluid as a medium to transfer 100–1000 MPa of high pressure, acting on quinoa seeds, so that some large molecular substances in quinoa seeds are changed.	(1) Nutrient loss and chemical composition changes are minimal; (2) Higher pressure induces complete starch gelatinization, though the retrogradation and textural properties of starch gels were little affected.	[82,83]
	Atmospheric pressure cold plasma (ACP)	Cold plasma flow is applied to the surface of quinoa seeds to achieve sterilization.	(1) Wide sterilization range; (2) Improving thermal stability and reduced gel strength.	[84]
	Sonication	Due to the cavitation effect, high shear forces, microjets, and shock waves are generated, which leads to the expansion of some protein structures in quinoa seeds and the disintegration of large aggregates.	Improved functional properties of quinoa seed protein.	[85]

## Data Availability

Data is contained within the article.

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
