# Peer review of "Research Progress of Quinoa Seeds (Chenopodium quinoa Wild.): Nutritional Components, Technological Treatment, and Application"

_foods, 2023, doi:10.3390/foods12102087_

Round 1

Reviewer 1 Report

1.     This review is an extensive study of the literature on a little-used pseudo-cereal, quinoa. It cannot however be termed a critical review, as it presents a vast number of literature “facts” without looking for, and suggesting resolution for, any conflicts between different reports. There is no such suggested resolution even when it is stated that reports are conflicting, e.g. lines 166-178. This alone is reason for its rejection; a review should be critical, not just a compilation from a literature search.

2.     Abstract: the statement “differs from the conventional cereals by the botanical traits” (which should be “in its botanical traits”) is meaningless without further details.

3.     The English can be understood but contains far too many serious syntactical errors, especially frequent incorrect usage of the definite article. It needs professional language editing. This is surprising because the corresponding author is from an institute where English is the working language, and he would only have to walk down the corridor to find a native speaker. It is essential that the English be brought up to the standard expected of the corresponding author’s institute.

4.     Line 32 “… including Europe, North America, Canada …”. This reviewer believes that Canada is in North America.

5.     Line 42: “… than other cereals.”. However, they have implied that quinoa is not really a cereal. This confusion appears throughout the manuscript.

6.     Line 150: “amylose of quinoa seeds appears to be more branched than the other chains”. To what does “other” refer? The authors do not seem to be aware that amylose is a glucose polymer, comprising mainly long (1→4)-α linked chains with a relatively small number of branch points made up of (1→6) -α linkages.

7.     Line 153 and elsewhere: use a proper micron symbol with Greek mu µ, not “um”.

8.     The bibliography does not follow the journal requirements for journal names (capitalization) or for spacing (e.g. line 161 there should be a space before the square opening  parenthesis.

9.     Line 161: what does “lower retrogradation percentage” mean? The authors need to look up how retrogradation is defined.

10.  Line 163: “speculated that the quinoa seed starch may supply as a suitable material for producing debranched starches that may further shed light on the utilization of quinoa seeds”. What is the reason and justification for this bold statement?

11.  Line 181: “varied proportions of starch-lipid complexes occur in the quinoa seeds”.  Under what circumstances do these variations occur? Between growth conditions, varieties, … ?

12.  Bibliography: it seems that the authors have not read the pdf file produced by the publisher, e.g. number 36 and many other places.

Author Response

(1)

  1. This review is an extensive study of the literature on a little-used pseudo-cereal, quinoa. It cannot however be termed a critical review, as it presents a vast number of literature “facts” without looking for, and suggesting resolution for, any conflicts between different reports. There is no such suggested resolution even when it is stated that reports are conflicting, e.g. lines 166-178. This alone is reason for its rejection; a review should be critical, not just a compilation from a literature search.

The manuscript was modified, Lines166-178 was modified, please refer to Lines 194-271; Also, the manuscript has been checked and corrected.

  1. Abstract: the statement “differs from the conventional cereals by the botanical traits” (which should be “in its botanical traits”) is meaningless without further details.

We're talking about quinoa as a pseudocereal because it is often mistaken for conventional cereals and the two are easily confused. Quinoa is also cooked in a similar way to wheat, barley and maize. However, in botanical terms, they belong to other plant families.

 We are sorry for the confused expression, the authors would like to explain that quinoa comes from Chenopodium family, while the conventional grains, such as wheat, oat, corn, etc., are from Grass family.

“The botanical traits” was changed into “The botanical terms”. Line 16.

  1. The English can be understood but contains far too many serious syntactical errors, especially frequent incorrect usage of the definite article. It needs professional language editing. This is surprising because the corresponding author is from an institute where English is the working language, and he would only have to walk down the corridor to find a native speaker. It is essential that the English be brought up to the standard expected of the corresponding author’s institute.

  The manuscript has been modified with the instruction of a native English speaker.

  1. Line 32 “… including Europe, North America, Canada …”. This reviewer believes that Canada is in North America.

    Modified. Please refer to Line 32.

  1. Line 42: “… than other cereals.”. However, they have implied that quinoa is not really a cereal. This confusion appears throughout the manuscript.

   Modified. “the other cereals” was corrected into “conventional cereals”; Please refer to Lines 15,20,51,243, and 607.

  1. Line 150: “amylose of quinoa seeds appears to be more branched than the other chains”. To what does “other” refer? The authors do not seem to be aware that amylose is a glucose polymer, comprising mainly long (1→4)-α linked chains with a relatively small number of branch points made up of (1→6) -α linkages.

We are sorry for the confusion. The author intended to demonstrate that the amylose of quinoa seed starch contains more short chains.

The modified description please refer to Lines 179-183, and the cited article please refer to ”Tang, H.; Watanabe, K.; Mitsunaga, T. Characterization of storage starches from quinoa, barley and adzuki seeds. Carbohydrate Polymers 2002, 49, 13-22”.

  1. Line 153 and elsewhere: use a proper micron symbol with Greek mu µ, not “um”.

Corrected. Line 184.

  1. The bibliography does not follow the journal requirements for journal names (capitalization) or for spacing (e.g. line 161 there should be a space before the square opening  parenthesis.

  Modified. Line 192. Also, the manuscript has been checked and corrected.

  1. Line 161: what does “lower retrogradation percentage” mean? The authors need to look up how retrogradation is defined.

 According to Li and Zhu(2018) (Li G, Zhu F. Quinoa starch: Structure, properties, and applications[J]. Carbohydrate polymers, 2018, 181: 851-861.) The retrogradation percentage (R%, the percentage ratio between the enthalpy change of retrogradation and gelatinization) of quinoa starch was lower than that of kañiwa, sorghum, millet, maize, and wheat, but higher than that of amaranth starch.

The phrases was modified, please refer to Lines 192-197.

starches, and was higher than that of amaranth starch We are sorry for the mistake. The author intended to describe from the reference that quinoa starch is prone to be gelatinized because of the amylopectin

  1. Line 163: “speculated that the quinoa seed starch may supply as a suitable material for producing debranched starches that may further shed light on the utilization of quinoa seeds”. What is the reason and justification for this bold statement?

  Modified. Lines 198-200.

  1. Line 181: “varied proportions of starch-lipid complexes occur in the quinoa seeds”.  Under what circumstances do these variations occur? Between growth conditions, varieties, … ?

  Deleted.

  1. Bibliography: it seems that the authors have not read the pdf file produced by the publisher, e.g. number 36 and many other places.

  Modified.

Reviewer 2 Report

The manuscript is interesting, but some of the suggestions mentioned here should be taken into account:

Line 38. To mention examples of foods products containing quinoa seeds

Line 42. Which polysaccharides are considered prebiotics in quinoa seeds? Indicate

Line 45-49. the information in the paragraph should be linked to the previous paragraph.

Line 102-105. Explain the importance of the water absorption capacity of quinoa seeds.

Line 156 and line 197. Remove the red dot at the end of the text.

Line 198. The word different is repeated. Replace.

Line 233. The word foun should be corrected.

In figure 1. It is recommended that the figures be of the same size.

Fig 2. Review the format of the figure

In thermal treatments, explain the rationale for the methods, for example, extrusion

Author Response

The manuscript is interesting, but some of the suggestions mentioned here should be taken into account:

Line 38. To mention examples of foods products containing quinoa seeds

Presented, please refer to Line 39.

Line 42. Which polysaccharides are considered prebiotics in quinoa seeds? Indicate

According to the observation of Zhu et al.(2020), it was indicated that compared with cereals, the composition of quinoa seed polysaccharides is more similar with that of fruits and vegetables, therefore, quinoa polysaccharides is considered prebiotic due to their capability of increasing beneficial bacteria growth.

Detailed information please refer to “Zhu, F. Dietary fiber polysaccharides of amaranth, buckwheat and quinoa grains: A review of chemical structure, biological functions and food uses. Carbohydrate Polymer 2020, 248, 116819, doi:10.1016/j.carbpol.2020.116819.”

Brief description was also provided, Lines 43-46.

Line 45-49. the information in the paragraph should be linked to the previous paragraph.

 Modified. The sentences that not relative to the paragraph was deleted.

Line 102-105. Explain the importance of the water absorption capacity of quinoa seeds.

   The water absorption capacity (WAC) of food products is an important parameter, as it mainly affects profitability and quality. Quinoa seeds protein with high WAC may be useful to the food industry by preventing water loss in breads and cakes and by increasing yields of cured sausages, canned fish and frozen products. Brief explanation was provided in Lines 135-138.

Line 156 and line 197. Remove the red dot at the end of the text.

 Modified. Please refer to Line 185 and Line 233.

Line 198. The word different is repeated. Replace.

  Differed was replaced by varied, please refer to Line 242.

Line 233. The word foun should be corrected.

  Corrected. The word “foun” was replaced by “present”, Line 282.

In figure 1. It is recommended that the figures be of the same size.

Modified

Fig 2. Review the format of the figure

   Modified.

In thermal treatments, explain the rationale for the methods, for example, extrusion

The rational for the methods were summarized in Table 3, please refer to Line 435.

Reviewer 3 Report

This manuscript is a summary and review of the research papers of the last 10 years on the ingredients, functional characteristics, and industrial application of quinoa seeds. The paper is systematically well written. However, it should be added that the contents of comparative components are presented and explained.

Specific comments are made as follows:

1. Line 29. Please write down what types of excellent nutritional components are available.

2. Line 60. Please present the wheat protein content and compare it with quinoa protein.

3. Lines 63-65. Present the prolamin and glutelin content of wheat protein and compare it to quinoa.

4. Lines 150-152. Indicate the grain granule size.

5. Lines 174-178. Give a number of digestibility and explain it.

6. Lines 178-182. Indicate and describe your GI and eGI values.

7. Lines 216-218. Present the lipid content of soy and corn and compare to quinoa.

8. Lines 260-262. Present the content of phenolic compounds in other cereals and compare them to quinoa.

9. Lines 298-301. Please provide the content and compare it.

10. Lines 319-332. Present and compare the Saponins content of different samples.

11. Lines 429-432. Please present and compare the content of Saponins in different samples.

12. Lines 430-451. Present the content and compare and analyze it. Please also explain why the content is different.

13. Lines 551-554. Present and compare the protein and amino acid content of milk. Please also explain why the content is different.

Author Response

This manuscript is a summary and review of the research papers of the last 10 years on the ingredients, functional characteristics, and industrial applications of quinoa seeds. The paper is systematically well written. However, it should be added that the contents of comparative components are presented and explained.

Specific comments are made as follows:

  1. Line 29. Please write down what types of excellent nutritional components are available.

  Quinoa seeds contains richer contents of protein, lipids, and ashes, and its amino acids composition is more balanced, please refer to Line 29, Lines 597-600.

  1. Line 60. Please present the wheat protein content and compare it with quinoa protein.

Presented. Please refer to Lines 66-67 and Table 2.

  1. Lines 63-65. Present the prolamin and glutelin content of wheat protein and compare it to quinoa.

Presented. Please refer to Lines 70-74

  1. Lines 150-152. Indicate the grain granule size.

Presented, please refer to Line 29.

  1. Lines 174-178. Give a number of digestibility and explain it.

Presented. Please refer to Lines 198-206, Lines 211-216.

  1. Lines 178-182. Indicate and describe your GI and eGI values.

Glycemic index (GI) describes the blood glucose response after consumption of carbohydrate-containing test food relative to a carbohydrate-containing reference food, typically glucose. According to the GI value, food can be divided into three different levels: low GI food (GI ≤ 55), middle GI food (56 ≤ GI ≤ 69) and high GI foods (GI ≥ 70) (FAO/WHO), The GI is generally expressed by the estimated glycemic index (eGI), which can be calculated based on the hydrolysis extent of a sample compared to the white bread.  Please refer to the literature of “Peng M, Yin L, Dong J, et al. Physicochemical characteristics and in vitro digestibility of starches from colored quinoa (Chenopodium quinoa) varieties[J]. Journal of Food Science, 2022, 87(5): 2147-2158.(Lines  218-226)

7.Lines 216-218. Present the lipid content of soy and corn and compare to quinoa.

The comparison of lipid contents and composition between quinoa and the conventional grains are presented in Lines 257-258, Lines 254-257, and Table 2.

  1. Lines 260-262. Present the content of phenolic compounds in other cereals and compare them to quinoa.

Presented. Please refer to Lines 275-277, Lines 343-344 and Table 2.

9.Lines 298-301. Please provide the content and compare it.

Modified. Please refer to Lines 348-356

10.Lines 319-332. Present and compare the Saponins content of different samples.

Presented, please refer to Lines 376-378 and Table 2.

  1. Lines 429-432. Please present and compare the content of Saponins in different samples.

The content of Lines 504-513 is related to the effect of high hydrostatic pressure on the properties of quinoa and grain starcher, the authors did not provide information on the variation of saponins.

  1. Lines 430-451. Present the content and compare and analyze it. Please also explain why the content is different.

 Modified and analyzed, please refer to Lines 483-495.

  1. Lines 551-554. Present and compare the protein and amino acid content of milk. Please also explain

Presented. Please refer to Lines 639-643.

Round 2

Reviewer 1 Report

1.     Stating “We are sorry to have confused you” at the start of each response is inappropriate, when, for example, the comment was suggesting that something needed to be added, and the response was to add it.

2.     There are still many English problems, e.g. “enable 34 it gains increasing popularity”, and “it differs from the conventional cereals by the botanical terms.”, which is incomprehensible. It was stated that it had been reviewed by a native speaker, but this was not done thoroughly. To repeat what was said earlier, the authors work in an institute where English is one of the working languages, so there is no excuse for this happening on a second review.

3.     Given this careless or neglect on the part of the authors to what was said on the first review, there is no point in this reviewer wasting time on reviewing further. 

Author Response

April 28, 2023

Dear Editor,

Thank reviewer’s comments for improving our manuscript. We have invited native-English expert to revise the manuscript based on previous twice review by native-English experts during our submission process.

The changed section and words are in red color.

I hope the revised manuscript will meet your request.

  Best regards

John Shi, PhD

Senior Scientist

Guelph Research and Development Center, AAFC
